# Ultrasonography and Postmortem Magnetic Resonance Imaging of Bilateral Ocular Disease in a Heifer

**Takeshi Tsuka** [1,*], **Yuji Sunden** [1]**, Takehito Morita** [1]**, Md Shafiqul Islam** [2,3] **and Osamu Yamato** [2]

1  Clinical Veterinary Sciences, Joint Department of Veterinary Medicine, Faculty of Agriculture, Tottori University, 4-101, Koyama-Minami, Tottori 680-8553, Japan; sunden@tottori-u.ac.jp (Y.S.); morita@tottori-u.ac.jp (T.M.)
2  Laboratory of Clinical Pathology, Joint Faculty of Veterinary Medicine, Kagoshima University, 1-21-24 Korimoto, Kagoshima 890-0065, Japan; si.mamun@ymail.com (M.S.I.); osam@vet.kagoshima-u.ac.jp (O.Y.)
3  Department of Pathology and Parasitology, Faculty of Veterinary Medicine, Chattogram Veterinary and Animal Sciences University, Khulshi, Chattogram 4225, Bangladesh
*  Correspondence: tsuka@tottori-u.ac.jp

**Abstract:** Bovine ocular diseases are typically characterized by the concurrent appearances of both macroscopic and intraocular abnormalities. This study examines the diagnostic efficacy of a combination of ultrasonography and magnetic resonance imaging (MRI) for the bilateral ocular disease observed in a 9-month-old Japanese Black heifer. This case presented with bilateral strabismus and a white-colored lens structure in the right eye. A combination of ultrasonography and MRI revealed formations of corn-like and V-shaped membranous structures within the vitreous cavities of the left and right eyeballs, respectively. In the right eye, a cataract was suspected on both ultrasonogram and MRI. This case involved bilateral retinal detachments and strabismus similar to the signs of an autosomal recessive hereditary ocular disease; however, the cataract in the right eye differed from that hereditary disease. Finally, in genetic analysis, a known mutation of the *WFDC1* gene was not detected. Ultrasonography is superior to MRI in demonstrating intraocular pathological changes. On the other hand, MRI is helpful for evaluating invasiveness of the ocular lesions to the peripheral structures. Thus, the combined use of these imaging modalities is recommended for diagnosing various bovine ocular diseases.

**Keywords:** cataract; heifer; magnetic resonance imaging; retinal detachment; ultrasonography





## 1. Introduction

Cattle are frequently observed with various ocular diseases classified into either acquired or congenital types. The acquired types are mostly associated with trauma inducing corneal damage and intraocular hemorrhage, and local or systemic infections inducing infectious bovine keratoconjunctivitis and endophthalmitis [1]. The congenital types are commonly characterized by multiple exhibitions of the macroscopic appearances and intraocular changes. In Japan, multiple ocular defects are known to be caused by an autosomal recessive hereditary disease in Japanese Black (JB) beef cattle (abbreviated as JB-MOD) [2,3]. The macroscopic appearances associated with bovine congenital ocular diseases include exophthalmos in anterior segment dysgenesis, small eyelids in microphakia or aphakia, cloudiness of the lenses in cataracts, bilateral heterochromia iridis associated with an MITF mutation, and bilateral strabismus in JB-MODs [3–13], but these are mostly not diagnostic. Bovine ocular diseases can be broadly differentiated based on a variety of concurrent patterns of intraocular abnormalities such as cortical and anterior and/or posterior subcapsular cataracts, retinal detachments, microphakia or aphakia, and persistent hyaloid artery [1,3,5–12]. Clinical use of imaging modalities for the bovine eyes can allow observations of the intraocular abnormalities.

Ultrasonography has previously been used to diagnose various ocular diseases in humans [14] and in animals such as dogs and cats [15–17], camels [18], sheep and goats [19], and deer [20]. Some bovine reports have used ocular ultrasonography for measurements of normal bovine ocular structures [21,22] and the diagnosis of ocular diseases [1,3–6]. Magnetic resonance imaging (MRI) usage has allowed for clear visualizations of the lesions within the retrobulbar regions as well as the eyeballs in human patients [23] and in canine, feline, and bovine cases [4,5,17,24,25], and has been used postmortem in the bovine cases [5,24]. Additionally, computed tomography (CT) has been frequently used for dogs and cats with various ocular diseases [16,17]. The bovine cases with JB-MODs have also been examined by CT together with ultrasonography [3]. This study includes the diagnostic efficacy of a combination of ultrasonography and MRI to assess the bilateral intraocular abnormalities, leading to the detection of the common and different findings against the characteristics of JB-MODs. Additionally, the results are discussed using previous human and animal reports on how to use these imaging modalities to differentiate between various bovine ocular diseases.

## 2. Case Presentation

A 9-month-old JB heifer presented with visual impairments in both eyes, soon after birth. The animal showed an obviously delayed growth of the body, developed gradually after transfer to a fattening farm at 8 months old, followed by difficulty in standing because of the extremely progressive weakness associated with continuous malnutrition due to defeat in competition between groups. In the left eye, the eyeball protruded slightly and involved a severe strabismus, resulting in the movement of half of the lens under the anterior aspect of the eyelid (Figure 1a). Pigmentation was evident in the bulbar conjunctiva. In the right eye, the eyeball was macroscopically normal in size, but involved a moderate strabismus (Figure 1b). The lens was seen as a whole, white-colored, and clouded structure.

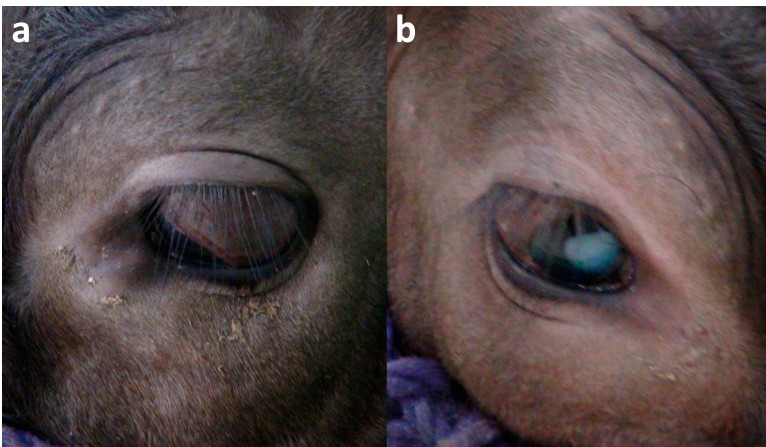

**Figure 1.** Macroscopic appearances of the left eye (**a**) and right eye (**b**). (**a**) Severe strabismus is evident in the slightly protruded left eye. (**b**) Moderate strabismus is evident in the right eye with a whole, white-colored lens, indicating a cataract.

Ophthalmic examination revealed no lesion within the surfaces of the cornea in either eye. No dazzle reflex and pupillary reflex appeared when bright lights were suddenly exposed to each eye.

Ocular ultrasonography was carried out using a portable-type ultrasound machine (MyLabOne VET, Esaote Co., Genoa, Italy). Prior to the examination, an ophthalmic local anesthetic (Benoxil ophthalmic solution 0.4%, Santen Pharmaceuticals Co., Ltd., Osaka, Japan) was topically applied for the surface of the cornea. A 10.0 MHz linear probe was applied to the surfaces of the cornea in the horizontal plane (medial to lateral direction) soon after topical use of an ultrasound gel, while the animal was kept in standing position without sedation. Ultrasonography of the left eyeball revealed that the anterior aqueous and vitreous humors were normally anechoic (Figure 2a). The anechoic lens was surrounded by echogenic lines of

the anterior and posterior lens capsules. A corn-like echogenic structure appeared between the posterior lens capsule and the deepest scleroretinal rim within the vitreous body when the probe was slightly moved medially from the center (Figure 2b). It was unclear whether the proximal region of this structure had adhered to the posterior lens capsule. The widest proximal region of the corn-like echogenic structure was approximately 1 cm thick. The distal region of this structure adhered to the deepest scleroretinal rim (possibly corresponding to the optic disc) and was thinnest at 2 to 3 mm thickness. A persistent hyaloid artery was suspected based on this ultrasonographic finding.

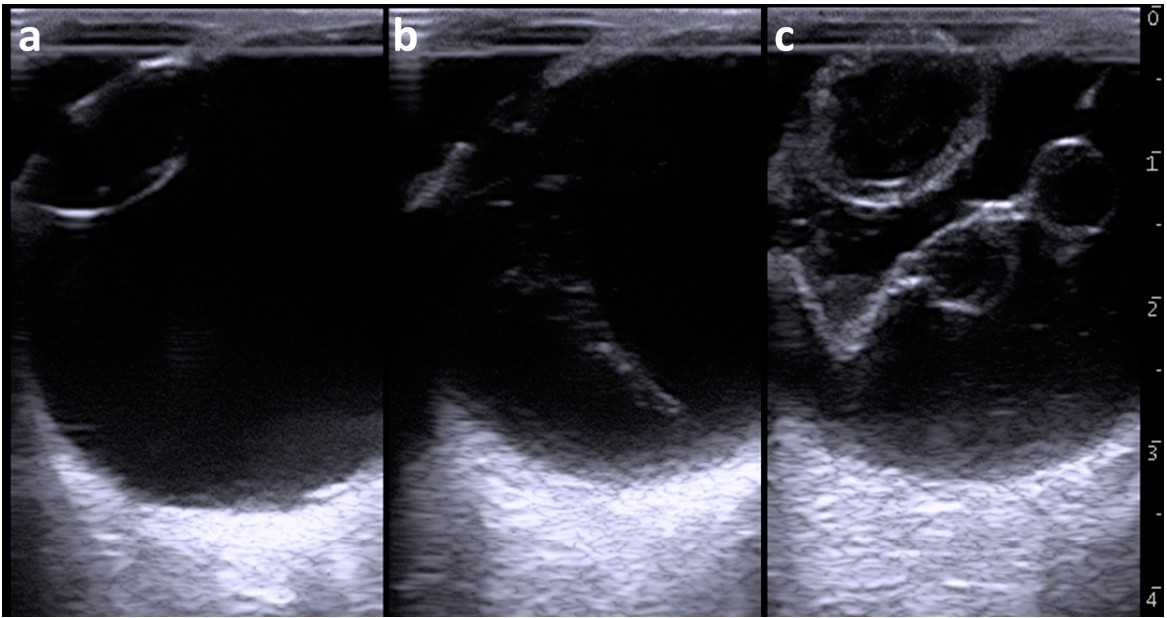

**Figure 2.** Ultrasonographic appearances of the left eyeball (**a**,**b**) and right eyeball (**c**). (**a**) The lens is anechoic and surrounded by echogenic lines of the anterior and posterior lens capsules within the left eyeball. (**b**) The corn-like structure is heterogeneously echogenic and present between the posterior lens capsule and the deepest scleroretinal rim within the vitreous body of the left eyeball. (**c**) A V-shaped membranous structure is present and accompanied by two cystic structures within the vitreous body of the right eyeball. The enlarged lens is heterogeneously anechoic to hypoechoic and is lined by the thickened and irregular anterior and posterior lens capsules. Scale = 10 mm.

Ultrasonography of the right eyeball revealed that the lens was lined by thickened and irregular anterior and posterior lens capsules (Figure 2c). The anterior lens capsule appeared as a heterogeneous and 3 to 4 mm thick structure and was slightly thicker compared with that of the posterior lens capsule (2 to 3 mm thickness). The posterior lens capsule appeared as a heterogeneous echogenic architecture lined by two hyperechoic anterior and posterior lines. The lens was heterogeneously seen with a mixture of anechoic and hypoechoic contents. The vitreous humor was heterogeneously anechoic and included hypoechoic contents. A V-shaped membranous structure could be seen within the vitreous body. The membranous structure was irregular and 1 to 3 mm thick. The tip of the V-shaped structure was located in the area of the optic disc, although it was unclear whether it ended within the optic disc. The two proximal edges of the V-shaped structure ended in the scleroretinal rim near the ciliary body. Two cystic structures were formed alongside each other in the center of one line of the V-shape of the membranous structure. The cystic structures were outlined by irregular and 1 to 2 mm thick echogenic lines and included heterogenous anechoic contents. The diameters of the two cystic lesions were 6.2 × 6.6 mm and 8.0 × 6.1 mm. No cystic lesion was evident in the other line of the V-shaped membranous structure.

This case was euthanized one day after the examination according to the owner's strong request because this case showed a moribund condition. At that time, MRI post-mortem examination was conducted for the skull that was obtained at necropsy using a low-field scanner (AIRIS Vento 0.3 T, Hitachi Medical Corporation, Tokyo, Japan) and a human knee coil. T1-weighted (time of repetition (TR), 450; time of echo (TE), 21; slice thickness, 5 mm), T2-weighted (TR, 3224; TE, 100; slice thickness, 5 mm), and fluid-attenuated inversion recovery (FLAIR; TR, 11,000; TE, 100; slice thickness, 5 mm) images were acquired. In the dorsal section of the left eye on the T1-weighted, T2-weighted, and FLAIR images, the corn-like structure could not be observed within the vitreous body (Figure 3).

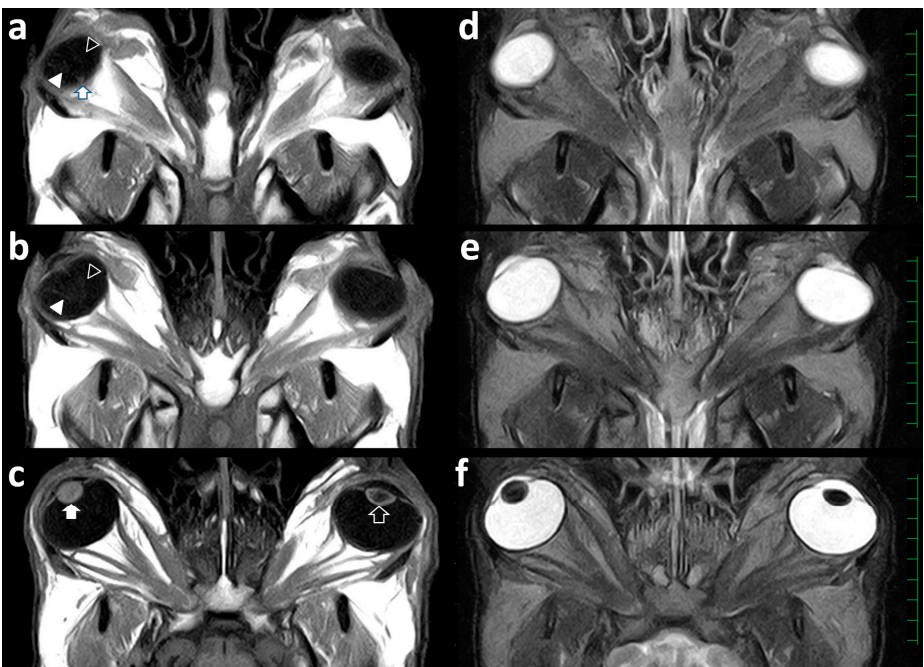

**Figure 3.** Dorsal T1-weighted (**a**–**c**) and T2-weighted (**d**–**f**) views of the skull demonstrating the left and right eyeballs. (**a**) A V-shaped structure (empty and filled arrowheads) is seen within the vitreous body of the right eyeball. The tip of the V-shaped structure ends in the area of the optic nerve (arrow). (**b**) Two cystic structures are slightly evident in the center of one line of the V-shaped structure (empty arrowhead), despite no cystic structure being evident in another line (filled arrowhead) within the vitreous body of the right eyeball. A corn-like structure is not evident within the vitreous body of the left eyeball. (**c**) The right lens (filled arrow) is a spherical structure appearing entirely by a high signal intensity. The left lens (empty arrow) is normally visualized as a low signal intensity's center surrounded by a high signal intensity's line of the anterior and posterior lens structures. (**d**–**f**) Abnormal membranous structures are not evident within the vitreous bodies of the left and right eyeballs. The right lens is enlarged in the anteroposterior direction. Scale = 10 mm.

Intraocular structures such as the lens, the ciliary body, and the anterior chamber were not abnormal in the left eye. The dorsal sections of the right eye on the T1-weighted (Figure 3a,b) and FLAIR images revealed a V-shaped line within the vitreous body. The tip of the V-shaped structure was located at the beginning of the optic nerve. The medial line of the V-shaped structure was irregularly thickened at the center (Figure 3b). On the T1-weighted dorsal section of the right eye (Figure 3c), the lens appeared heterogeneously as a high signal intensity and was slightly thicker in the anteroposterior direction compared with that of the left eye. On the T1-weighted images, it was not possible to differentiate the border between the inner content of the lens and the anterior and posterior lens capsules. The lens in the right eye was heterogeneously seen as a low signal intensity on the T2-weighted (Figure 3f) and FLAIR images. There was no abnormality of the optic nerve,

the extraocular muscles, or the fat tissues within the retrobulbar regions of both orbits. Additionally, there was no abnormality in the brain structures.

At necropsy, there were no macroscopic abnormalities in the thoracic or abdominal organs. The sizes of both eyes were normal, but the corneas were cloudy. In the cut surfaces of both eyes, the hyaloid artery was not visible within the vitreous bodies. Membranous structures were present within the vitreous cavities of both eyes and were characterized as translucent and white-colored. Additionally, the membranous structures were partly cystic or coil form. In the right eye, the membranous structure appeared to bridge between the optic disc and the scleroretinal rim close to the ciliary body. The lens of the right eye appeared as a spherical shape and was diffusely clouded (Figure 4a).

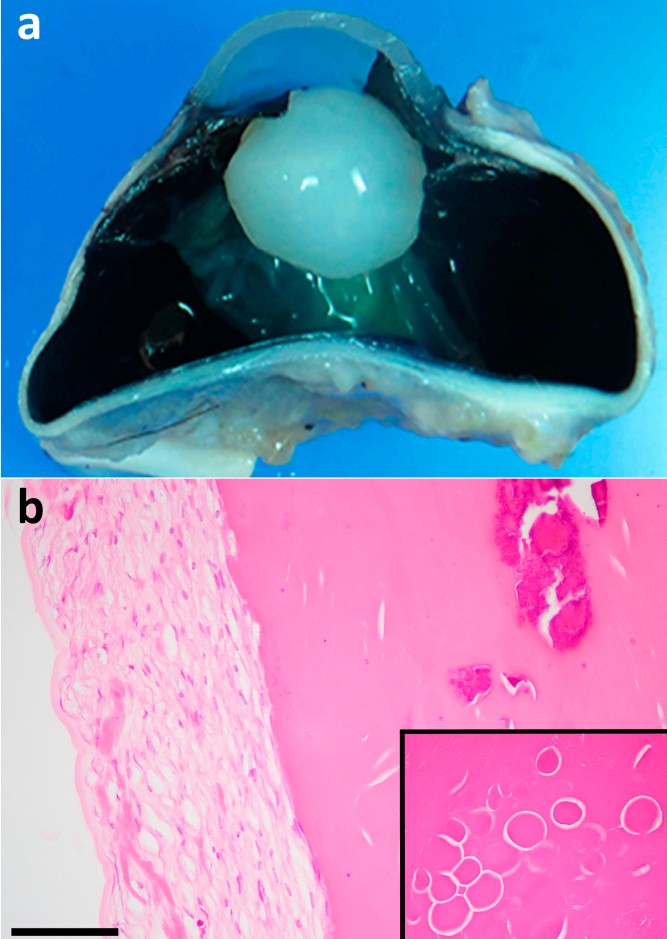

**Figure 4.** Macroscopic view (**a**) and histopathology (**b**) of the lens within the right eyeball. (**a**) The lens is spherical, and diffusely clouded and white-colored. (**b**) In the cortex of the lens, mineralization (right upper area) and aggregated globular bodies (Morganian globules; inset) are scattered. The subcapsular region (left area) contains a mild proliferation of fibrous cells with an eosinophilic collagenous fibers deposition on the entire circumference of the lens structure (HE). Bar = 100 µm.

Both eyeballs were fixed by formalin and routinely processed for paraffin-embedding and staining of sections by H&E. Histopathologically, the right lens was degenerated and showed the formation of globular aggregates (Morganian globules) and calcification in the cortex (Figure 4b). Additionally, spindle-shaped fibrous cells were proliferated in the whole circumference of the lens capsule and were especially marked in the anterior capsule of the lens compared with the posterior capsule of the lens. These findings indicated the formation of a cortical cataract in the right eye; the lens structures did not include proliferation of the lens epithelial cells as the histopathological characteristics of the anterior and posterior subcapsular cataracts. The left lens showed no histopathological abnormality.

The membranous structures within both eyeballs were consistent with a detached retina, characterized by the photoreceptor layer to the optic nerve layer (Figure 5a,b), suggesting that the structures had torn from the retinal pigment epithelial cell layers. The detached retina maintained its layered structures but showed mild degeneration and atrophy. Furthermore, the detached retinas sometimes showed distorted or cystic changes. Another histopathological change was the accumulation of inflammatory cells, mainly lymphocytes and macrophages, in the sclera and the base of the cornea.

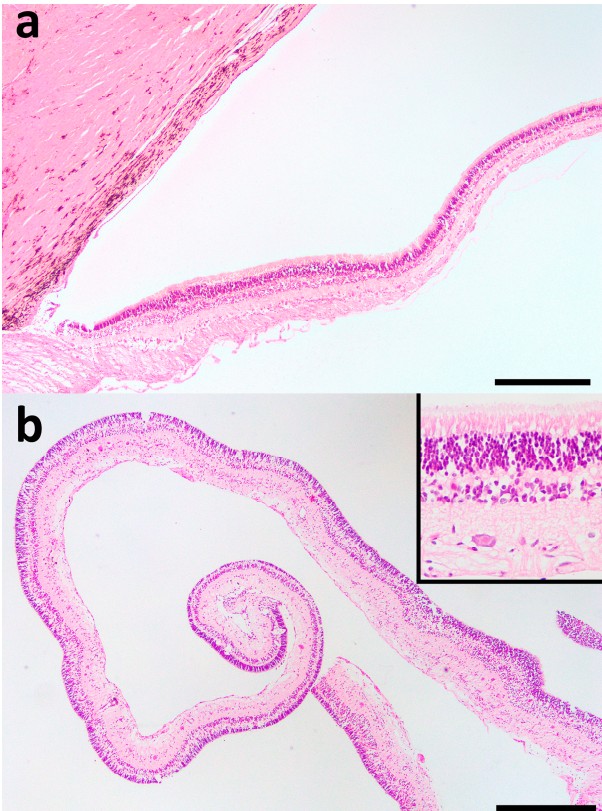

**Figure 5.** Histopathology of the membranous substances within the left eyeball (**a**) and right eyeball (**b**). (**a**) Membranous structures within the left eyeball consist of the retina. The retina is detached from the pigment layer (left upper area). The optic disc is located in the left lower area. (**b**) Distorted membranous structures consist of the retina within the right eyeball. The retinae of both sides are atrophic; however, the layered construction is recognizable including a photoreceptor layer to the optic nerve layer as shown in the inset (HE). Bar = 500 μm.

Polymerase chain reaction (PCR) was performed to amplify the targeted area including g.10567100_10567101 from the bovine genome database (ARS-UCD1.2), which is the location of the single nucleotide insertion mutation (*WFDC1*:c.198_199insC) previously identified for multiple ocular defects in JB calves [2]. Amplifications were performed by forward (5′-TAGGCGGAGGAGGTAGGC-3′) and reverse (5′-CAGCCGTTGTAGCAGCAGC-3′) primers designed based on the sequence of bovine *WFDC1* gene exon 2 (NCBI Reference Sequence NC_037345.1). The sequencing of the amplified band was performed by Sanger sequencing (Kazusa Genome Technologies Inc., Kisarazu, Japan). In PCR examination, the sequence of this case was homozygous wild-type in which a c.198_199insC mutation was not detected (Figure 6).

Based on the present results from these examinations, a congenital multiple ocular disease was considered in this case, despite the fact that it did not correspond completely with any bovine ocular diseases (including a JB-MOD) reported previously.

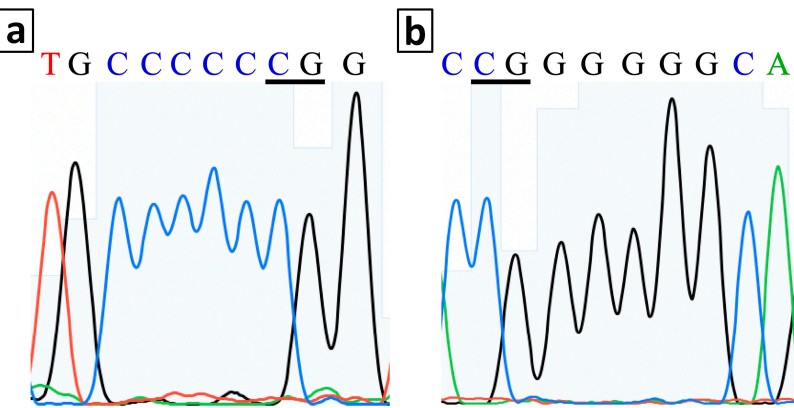

**Figure 6.** Forward (**a**) and reverse (**b**) sequences of the bovine *WFDC1* gene exon 2. Underlines show the wild-type nucleotide sequences at the position of g.10567100_10567101 in the bovine genome database (ARS-UCD1.2).

## 3. Discussion

The ocular abnormality in this case included strabismus macroscopically and retinal detachments ultrasonographically in both eyes. These findings were very similar to the ocular characteristics of JB-MODs [2,3]. However, the left eye of this case involved a cortical cataract, which is not included in the typical intraocular lesions of JB-MODs [3]. Additionally, the decreased size of the eyeball was also not evident in this case, despite it being typical in JB-MODs [3]. The difference in these ocular findings could be significant evidence to distinguish this case from JB-MODs. There are multiple types of bovine congenital ocular abnormalities apart from JB-MODs, in which the findings are similar to those in this case [5,7,8]. On the other hand, based on the previous human reports describing co-expression of multiple genes including the *WFDC1* gene and the association of a *CEP290* gene mutation for congenital retinal diseases [26,27], the mutations in the other genes as well as an unknown region of *WFDC1* gene may cause JB-MODs.

Congenital cataracts are believed to be inherited as a simple autosomal recessive trait in Jersey and Holstein Friesian species [6,7], but they can be associated with the provocative factors of ocular developmental abnormalities such as low levels of antioxidants (selenium and vitamin E) in mother cows and in utero infections of bovine viral diarrhea virus [8,13]. Congenital cataracts are commonly found bilaterally in cattle [6,7], but have been reported unilaterally in a deer [20], and, respectively, unilaterally and bilaterally in three and five of eight camels [18]. Unilateral cataracts are common in humans [20]. The cataract affecting unilaterally in this case might be atypical in bovine congenital cataracts or might be an acquired type. Bovine congenital cataracts commonly accompany concurrent intraocular abnormalities such as retinal detachment, irideremia, microphakia, and lens luxation [7]. Thus, the concurrent involvements of both retinal detachment and cataract suggested a congenital type in this case. Additionally, a congenital cortical cataract was considered as the lens lesion in this case, because the affected cattle can present with any common type of congenital cataracts such as mature cortical cataracts, nuclear cataracts, and anterior and/or posterior subcapsular cataracts [6–8]. However, it is difficult to determine the specific disease name (such as cataract) based on one lesion focused from mixtures of multiple ocular lesions within an affected eye, and which are typically found in multiple bovine congenital ocular abnormalities; some of which are called by different names and may be the same disease.

Congenital ocular abnormalities associated with maternal hypovitaminosis A have included bilateral ocular dysplasia such as hypogenesia of the optic nerve, retina atrophy, aphakia, and the absence of a vitreous body, but not cataracts [8,12]. Congenital microphthalmia can occur due to abnormal developments of the optic vesicle or failures in the establishment of intraocular pressure contributing to normal growth of the eyeball [5]. Congenital microphthalmias are partly associated with gene mutations in the MITF

gene of Holstein and German Fleckvieh breeds [10,11] and accompany other intraocular abnormalities such as aphakia, microphakia, and retinal dysplasia [5]. The intraocular findings are superior to macroscopic appearances such as the size of eyes, the degree of cloudiness of the eyes, and the unilateral or bilateral affections for clinical significance in differentiating bovine ocular diseases. The use of imaging modalities can allow for effective visualizations of the intraocular structures compared with the macroscopic observations and have previously been reported in bovine cases [3,5,24].

Ultrasonography has previously been used for the eyes of dogs, cats, cattle, camels, sheep, goats, and deer with various ocular diseases [1,3–6,16–20]. The ultrasound frequency of the probes is recommended as 7.5 to 10 MHz for ocular ultrasonography in veterinary medicine [17]. A 7.5 MHz or 10 MHz probe was used for the eyes of small and large ruminants [4,5,18,19,21,22]. Additionally, the ultrasound probes were placed either on the cornea surfaces with instillation of a topical anesthesia [1,4,5,18–20] or in contact with eyelids [21,22] for ruminants examined under sedation [5,18,19] or without sedation [1,4,20–22]. Ocular ultrasonography should be performed for non-sedated cattle despite being restrained in a crush or stocks, because sedation can cause rotation of the globe and elevation of the nictitating membrane [17].

On the ultrasonogram of bovine eyes, the intraocular structures can be clearly identified by the different echogenicity of each structure; the cornea, the anterior and posterior lens capsules, and the scleroretinal rim appear hyperechoic; and the aqueous humors, the lens, and the vitreous humors appear anechoic [21]. There were some bovine reports describing the clinical efficacies of ocular ultrasonography for observations of intraocular abnormalities such as aphakia, microphakia, absence of the lens, retinal detachment, and a persistent hyaloid artery [1,3–6]. Retinal detachment was one of the intraocular abnormalities of bovine endophthalmitis and JB-MODs when using ultrasonography [1,3]. Additionally, the bilateral strands within the vitreous fluid might appear as retinal detachment, as demonstrated on the ultrasonogram when scanning the affected eye of a bovine case with congenital cataract [6]. Retinal detachment can appear ultrasonographically as a V-shaped and curvilinear membranous structure attached to the optic disc for the complete types and as a partial tear of the retina lifted from the posterior wall of the eyeball for the partial types [1,17]. Ultrasonography of the right eyeball in this case identified the complete type of retinal detachment. However, to our best knowledge, there was no previous human or veterinary report showing a cystic lesion formed in the detached retina, which was ultrasonographically visualized within the right eyeball and was identified within the left eyeball at autopsy, although the left lesions were not detected ultrasonographically. The cystic lesions were histopathologically derived from the detached retina itself, but the etiology of the lesions was not determined. On the ultrasonogram of the left eyeball, the retinal detachment was misdiagnosed as a persistent hyaloid artery based on the ultrasonographic appearance of a corn-like structure in the center of the vitreous body of the left eye, because of a strong similarity to the ultrasonographic appearances in the canine and deer cases with persistent hyaloid arteries [17,20]. A retinal detachment commonly advances through the following three phases if left untreated in human patients: (1) the retinas tear completely or partially tear from the sclera, and the subretinal fluids accumulate beneath the detached retinas; (2) the detached retinas gather into a cord-like structure due to the extensive retraction in the center of the vitreous body; and (3) the globes are finally shrunken [14]. The retinal detachment within the left eyeball might have been equivalent to the second phase when examined by ultrasonography. Although Doppler ultrasonography has not been applied for this case, it has previously been used to evaluate the vasculature in the bovine ocular structures [22] and may allow differentiation between the second phase retinal detachment and the persistent hyaloid artery, because blood flow was observed in some of the persistent hyaloid arteries, but not in the detached retinas [16].

The echogenicity of the cataractous lens varied between hypoechoic and hyperechoic in the internal structures and was surrounded by the curvilinear hyperechoic lines of the anterior and posterior capsules of the lens with varied thickness [17,18]. The echogenic distri-

bution of cataractous lenses is likely associated with the degree of maturity of cataracts [17]. Hypermature cataracts appear with increased echogenicity of the lens structures with irregular lenticular borders [19]. This case involved a cortical cataract that appeared as a homogenous anechoic internal structure within the hyperechoic lens capsule; this finding was similar to that of cortical cataracts [19,20].

MRI has previously been applied to the skulls of two autopsied bovine cases, allowing the detection of bilateral microphthalmia and an abnormal shape of the globes in one case, and multiple formations of masses extending from the cavernous sinus resulting in injury of the cranial nerves in another case with cavernous sinus syndrome [5,24]. Recently, MRI could provide diagnostic evidence such as the defect of the lens structure within the extended eyeball, when examined for the anesthetized bovine case with anterior segment dysgenesis [4]. These previous results indicates that high quality images are helpful for identifying the pathological morphology of the affected eyeballs themselves, and evaluating their invasiveness toward and/or from the retrobulbar and skull's structures, regardless of whether MRI is used antemortem or postmortem [4,5,24]. However, a discrepancy between the MRI and necropsy findings was also suggested when using MRI postmortem [5]. On the MRIs of both eyes in this case, the V-shaped structure of the detached retina and the cystic retinal lesions could barely be clarified within the right eyeball, but an atypical corn-like structure of the detached retina was not evident within the left eyeball. The low-field MRI scanner (0.3 Tesla) that was used for this case had limitations for visibility of the smaller structures compared with a high-field MRI scanner [23,25,28]. In this case, the cataractous lens appeared as a high signal intensity and a heterogeneous low signal intensity on the T1-weighted and T2-weighted images of the right eyeball, respectively. The MRIs of the right lens structure were recognizable from the normal signal intensities in the left lens structure but are unlikely to be similar to the signal intensities of the MRIs in human cases with cataracts [23]. The signal intensities of the cataractous lenses in the acquired types varied on T1-weighted images and were high on the T2-weighted images; this might have been attributable to the increased hydration [23]. On the other hand, there were no abnormalities in the signal intensities of T1-weighted and T2-weighted images of the congenital cataracts [23]. The different patterns of the signal intensities in the MRIs of cataracts may be associated with the differences in the internal substances accumulated within the cataractous lenses between human and bovine cases. Contrast MRIs can lead to different enhancements related to each ocular structure, facilitating the clear anatomical identification of the neighbor structures of the lens; this could be utilized for evaluating the cataracts [25]. Contrast MRIs can be useful for diagnosing bovine cataracts, if examined antemortem, because the effectiveness of this technique has already been confirmed for bovine cases [29]. Elevated diagnostic efficacies of MRIs for bovine ocular diseases are required by establishing age-matched or breed-matched indexes of bovine ocular structures such as basic MRI findings and normal measurement values [5].

To compare between the quality of image obtained from the use of ultrasonography and MRI in this case, ocular ultrasonography seems to be superior to MRI in the clear identification of intraocular soft tissue lesions. On the other hand, with ocular ultrasonography, it is very difficult to demonstrate formation of the lesions within the deeper anatomical places including the retrobulbar region and the nerve tract between the optic nerve and brain. In this case, the combined use of ultrasonography and MRI was helpful for evaluating the lesion localized to the eyes within the skull, because it can compensate for shortcomings of these imaging modalities. Additionally, CT is superior to both ultrasonography and MRI for diagnosing the osseous lesions associated with ocular diseases when using a three-dimensional reconstruction [17,25]. The postmortem use of CT for the skulls of autopsied animals with JB-MODs enabled the visualization of intraocular abnormalities such as microphthalmia, retinal detachment, and persistent hyaloid arteries [3]. However, the degrees of clearness in the CT imaging of intraocular appearances have not been clarified because no images have been published [3]. CT is commonly inferior to MRI for the detection of the intraocular structures, because it provides a comparatively lower imaging resolution than

MRI, especially in reconstructed CT images [14,17,25]. The use of CT for a canine case with a persistent hyaloid artery failed to show a linear structure that could have been visualized on the ultrasonogram [16].

Various types of bovine multiple ocular diseases can cause visual impairments despite rarely being a cause of systemic signs [3,5,7–9]. These diseases may be overlooked by farmers, because they rarely threaten the survival of the affected animals. However, based on the results of this case, visual impairment can contribute to decreased body condition and a delay of growth due to indirect influences, such as a defeat in intra-herd competition. Furthermore, earlier clinical detection of these diseases can provide significant information for reproductive management, because optimal mating can prevent herd-level re-occurrence of the same ocular diseases, based on prior evidence about the carriers of causative genes [2,3,10,11]. Ultrasonography is superior to MRI and CT for earlier detection in the field, because it can be used for bovine eyes by simple operation and provides real-time imaging. MRI and CT can be used as a complementary imaging modality to provide significant evidence that cannot be obtained from ocular ultrasonography. Furthermore, MRI and CT may allow assessments from a forensic viewpoint, common in human medicine [28], when applied to the autopsied materials. Use of these imaging modalities for the eyes of affected bovine cases enables the assessment of patterns in intraocular abnormalities corresponding to each previous congenital or acquired ocular disease. In this study, the combined use of ultrasonography and MRI for this case could not identify a complete correspondence with those in previous bovine ocular diseases, including a JB-MOD, which was suspected before the examinations. Through the examination process, although no definitive diagnostic evidence was acquired, a JB-MOD could at least be denied.

### 4. Conclusions

Ultrasonography can be routinely used for early detection of bovine ocular diseases in the field. MRI can provide not only diagnostic, when used antemortem, but also forensic roles when used postmortem (as commonly seen in human medicine). The clinical use of these imaging modalities can contribute to determining the etiology of bovine ocular diseases. Additionally, the accumulation of imaging data obtained from the combined use of imaging modalities must be helpful for further detection of unknown congenital or hereditary bovine ocular diseases.

**Author Contributions:** Conceptualization, T.T.; methodology, T.T.; investigation, T.T., Y.S., T.M., M.S.I. and O.Y.; writing—review and editing, T.T. All authors have read and agreed to the published version of the manuscript.

**Funding:** This research received no external funding.

**Institutional Review Board Statement:** The study protocol was performed in accordance with the guideline of the institutional animal care and use committee of Tottori University.

**Informed Consent Statement:** We obtained informed written consent from the owner, which covered the usefulness of ultrasonography and MRI in differential diagnosis of this ocular disease from JB-MOD. Additionally, the owner agreed with the informed written consent about the meaningfulness of necropsy examination in order to clarify the reasons for the decreased body condition and the delay of growth as well as definitive diagnosis in this case.

**Data Availability Statement:** The data presented in this study are available on request from the corresponding author.

**Conflicts of Interest:** The authors declare no conflicts of interest.

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
