# Peer review of "Ultrasonography and Postmortem Magnetic Resonance Imaging of Bilateral Ocular Disease in a Heifer"

_ruminants, doi:10.3390/ruminants4010008_

Round 1

Reviewer 1 Report

Comments and Suggestions for Authors

Dear authors

The case report is nicely presented and clinical findings are interesting. I have few comments:

-title should include postmortem MRI, or clarify this was done after euthanasia, it is misleading otherwise.

Line 22-23: you mention no mutation found, then "from genetic analysis" which is redundant.

Line 62: "opthalmic exam of the bovine eyes" it is also redundant.

Lines 101-104. Was the animal seen because of the body condition? or because of the eye appearance? what was the body condition? why was the animal not examined earlier? As the animal had retinal detachment, was the animal blind then?

Line 110-112 "whole white colored lens" wouldnt it just called it a Cataract? which degree?

Line 114: normal not normally

Line 115: Measurement of the IOP showed...

Line 226 and discussion: Is there the possibility of another mutation? it is not clear what sequence is normal even with the underlines. The disease is similar to the one described but could it be another mutation? or how do you explain the differences on clinical signs? maybe due to the chronic state of the pathology?

Comments on the Quality of English Language

I have stated above some comments.

Author Response

The case report is nicely presented and clinical findings are interesting. I have few comments:

Thank you for your delighted comments. In the revised version, the corrected parts are highlighted by yellow boxes.

Question: Title should include postmortem MRI, or clarify this was done after euthanasia, it is misleading otherwise.

Answer: In the revised version, the title is changed as “Ultrasonography and postmortem magnetic resonance imaging of bilateral ocular disease in a heifer”.

Question: Line 22-23: you mention no mutation found, then "from genetic analysis" which is redundant.

Answer: We agree that the sentence in lines 22-23 is difficult to be understood. Thus, this sentence is corrected.

Question; Line 62: "opthalmic exam of the bovine eyes" it is also redundant.

Answer: According to this suggestion and the comment from the other reviewer, section of “Materials and Methods” is deleted by transfer of the descriptions of ocular ultrasonography, MRI and PCR examination into the section “Case presentation”.

Question: Lines 101-104. Was the animal seen because of the body condition? or because of the eye appearance? what was the body condition? why was the animal not examined earlier? As the animal had retinal detachment, was the animal blind then?

Answer: In this sentence, there was lack of explanation. An owner confirmed that the animal was blind at birth. However, in the newborn period, it was unknown whether the retinal detachment has been the cause of the visual impairment, because this case has not been examined by ocular ultrasonography in that time. In this case, transfer to a fattening farm at 8 months old was suspected to be a trigger on exhibitions of the delayed growth and difficulty in standing. This case had continuous malnutrition due to defeat in competition between groups because of its visual impairment. In the revised version, this sentence is corrected.

Question: Line 110-112 "whole white colored lens" wouldnt it just called it a Cataract? which degree?

Answer: On photo, the macroscopic view of "whole white colored lens" indicated a cataract. Thus, the legend of Figure 1 is slightly corrected.

Question: Line 114: normal not normally

Answer: According to the comment, this sentence is corrected.

Question: Line 115: Measurement of the IOP showed...

Answer: In the revised version, this sentence is deleted.

Question: Line 226 and discussion: Is there the possibility of another mutation? it is not clear what sequence is normal even with the underlines. The disease is similar to the one described but could it be another mutation? or how do you explain the differences on clinical signs? maybe due to the chronic state of the pathology?

Answer: We agree that the association of another mutation of WFDC1 gene. Our genetic investigation has been performed only for a known region of WFDC1 gene. Although in JB-MOD, only this mutation of WFDC1 gene was known previously, genetic associations of multiple genes including WFDC1 gene are reported for human patients with various congenital ocular diseases. Thus, in the revised version, the discussion about genetic factors of JB-MODs is added by the addition of two human reports as reference papers.

Reviewer 2 Report

Comments and Suggestions for Authors

The article is interesting mainly due to the correlation of ultrasound images with macroscopic findings. Suggestions:

Line 51  e 52    " regions as well as the eyeballs in human patients [22] and  in canine, feline, and bovine cases [4,5,16,23,24] as well as postmortem uses in the bovine  cases , I suggest changing to “and postmortem uses……”

Line 160           “The dorsal sections of the right eye on the T1-weighted and FLAIR images revealed a V-shaped line within the vitreous body (Figures 3a,b), change:

                           The dorsal 160 sections of the right eye on the T1-weighted (Figures 3a,b). and FLAIR images revealed a V-shaped line  within the vitreous body.

Figure 3            Figure 3 is indicated in the text in the wrong places.....

.Line 160  In Figure 3 there was no FLAIR image and more than once figure 3 is indicated after the word FLAIR.

. Line 160“The dorsal sections of the right eye on the T1-weighted and FLAIR images revealed a V-shaped line within the vitreous body (Figures 3a,b), change

                           The dorsal sections of the right eye on the T1-weighted (Figures 3a,b) and FLAIR images revealed a V-shaped line  within the vitreous body.

                           . Line 164  “On the T1-weighted dorsal section of the right eye, the 164 lens appeared heterogeneously as a high signal intensity and was slightly thicker in the anteroposterior direction compared with that of the left eye (Figure 3c).”change:

                            On the T1-weighted dorsal section of the right eye (Figure 3c), the  lens appeared heterogeneously as a high signal intensity and was slightly thicker in the anteroposterior direction compared with that of the left eye.

                           . Lines 168-170 “ The lens in the right eye was heterogeneously seen as a low signal intensity on the T2-weighted and FLAIR images (Figure 169 3f).Change:

                           Sugestão: “ The lens in the right eye was heterogeneously seen as a low signal intensity on the T2-weighted (Figure 169 3f) and FLAIR images.

                           .Figure 3  I suggest changing the images in figure 3, zoom on the T1 and FLAIR image of the right eye with a V-shaped structure.

                           . “However, the left???(right??) eye of this case involved a cortical cataract, which is not included in the typical intraocular lesions of JB-MODs “.

Author Response

The article is interesting mainly due to the correlation of ultrasound images with macroscopic findings. Suggestions:

Thank you for your providing kindly comments. The corrected parts are highlighted bu yellow boxes.

Question: Line 51-52 " regions as well as the eyeballs in human patients [22] and in canine, feline, and bovine cases [4,5,16,23,24] as well as postmortem uses in the bovine cases , I suggest changing to “and postmortem uses……”

Answer: According to this comment, this sentence is corrected.

Question: Line 160 “The dorsal sections of the right eye on the T1-weighted and FLAIR images revealed a V-shaped line within the vitreous body (Figures 3a,b), change:

The dorsal sections of the right eye on the T1-weighted (Figures 3a,b). and FLAIR images revealed a V-shaped line within the vitreous body.

Answer: According to this comment, this sentence is corrected.

Question: Figure 3 is indicated in the text in the wrong places.....

Answer: The place of Figure 3 is changed according to this comment.

Question: Line 160 In Figure 3 there was no FLAIR image and more than once figure 3 is indicated after the word FLAIR.

Answer: The place of Figure 3 is changed according to this comment.

Question: Line 164 “On the T1-weighted dorsal section of the right eye, the lens appeared heterogeneously as a high signal intensity and was slightly thicker in the anteroposterior direction compared with that of the left eye (Figure 3c).”change:

On the T1-weighted dorsal section of the right eye (Figure 3c), the lens appeared heterogeneously as a high signal intensity and was slightly thicker in the anteroposterior direction compared with that of the left eye.

Answer: According to this comment, this sentence is corrected.

Question: Lines 168-170 “The lens in the right eye was heterogeneously seen as a low signal intensity on the T2-weighted and FLAIR images (Figure 3f). Change:

“The lens in the right eye was heterogeneously seen as a low signal intensity on the T2-weighted (Figure 3f) and FLAIR images.

Question: Figure 3 I suggest changing the images in figure 3, zoom on the T1 and FLAIR image of the right eye with a V-shaped structure. “However, the left???(right??) eye of this case involved a cortical cataract, which is not included in the typical intraocular lesions of JB-MODs“.

Answer: Unfortunately, the FLAIR images did not have high quality of image applicable for journal’s publication, although could allow evaluation of the signal intensity of the eyes. In this case, imaging diagnosis of cortical cataract, as an uncommon intraocular lesion of JB-MODs, suggested involvement of ocular disease other than JB-MOD, supported by genetic analysis revealing no c.198_199insC mutation in WFDC1 gene. However, our results might not allow completely negation of hereditary ocular disease, because mutations in the other gene and an unknown region of WFDC1 gene might have been presented in this case. In the revised version, the new discussion is added.

Reviewer 3 Report

Comments and Suggestions for Authors

While the paper addresses the diagnostic efficacy of ultrasonography and magnetic resonance imaging (MRI) in a case of bilateral ocular disease in a 9-month-old Japanese Black heifer, there are several areas that could be improved:

The introduction could provide a clearer background on common bovine ocular diseases before delving into the specific case study. This context would help readers unfamiliar with the subject matter to better understand the significance of the diagnostic techniques used.

The description of the case lacks certain essential details, such as the duration of the observed symptoms, any relevant medical history, or the specific clinical manifestations of bilateral strabismus. Providing a more comprehensive overview of the case would enhance the reader's understanding.

he paper briefly mentions the combination of ultrasonography and MRI but lacks a detailed explanation of why this combination was chosen and how each modality contributes to the overall diagnostic process

it would be beneficial to include a more detailed comparative analysis, discussing the strengths and limitations of each imaging modality in the context of bovine ocular diseases.

The conclusion could be strengthened by suggesting potential avenues for future research

Ensure that in-text citations are consistently formatted, and include a complete list of references at the end of the paper to provide readers with the opportunity to explore related literature.

Author Response

While the paper addresses the diagnostic efficacy of ultrasonography and magnetic resonance imaging (MRI) in a case of bilateral ocular disease in a 9-month-old Japanese Black heifer, there are several areas that could be improved:

Thank you for your kindly suggestion. The corrected parts are highlighted by yellow boxes.

Question: The introduction could provide a clearer background on common bovine ocular diseases before delving into the specific case study. This context would help readers unfamiliar with the subject matter to better understand the significance of the diagnostic techniques used.

Answer: Thank you for your providing this comment.

Question: The description of the case lacks certain essential details, such as the duration of the observed symptoms, any relevant medical history, or the specific clinical manifestations of bilateral strabismus. Providing a more comprehensive overview of the case would enhance the reader's understanding.

Answer: In terms of your comment “the duration of the observed symptoms”, this case exhibited the gradually progressive symptom between 1 month after transfer to a fattening farm (at 8 months old). In terms of your comment “any relevant medical history”, this case has not been treated without medical request from the owner to veterinarians. In terms of your comment “the specific clinical manifestations of bilateral strabismus”, the owner has not noticed this ocular symptom, although the owner has noticed visual impairment when bought this case. Thus, with reference from the other reviewer’s comment, the description of this case’s information is corrected.

Question: The paper briefly mentions the combination of ultrasonography and MRI but lacks a detailed explanation of why this combination was chosen and how each modality contributes to the overall diagnostic process.

Answer: We think that ultrasonography is superior to MRI in quality of image of various soft tissue lesions within the affected eye, in addition with the real-time observation for unanesthetized animals. However, the use of ultrasonography is very difficult to appear the osseous lesions around the eye, retrobulbar lesions, the nerve tract lesions between the optic nerve and brain, and the cephalic lesions. Because we have suspected the affection of the lesions formed in the anatomical place in which could not be identified with ocular ultrasonography, postmortem MRI was carried out for this case. In the revised version, the description of discussion about above-mentioned contents is added.

Question: it would be beneficial to include a more detailed comparative analysis, discussing the strengths and limitations of each imaging modality in the context of bovine ocular diseases.

Answer: In terms of limitation of ocular ultrasonography and meaningfulness of combination use of ultrasonography and MRI, the description of discussion is added in the revised version.

Question: The conclusion could be strengthened by suggesting potential avenues for future research.

Answer: According to this suggestion, the description of conclusion is added.

Question: Ensure that in-text citations are consistently formatted, and include a complete list of references at the end of the paper to provide readers with the opportunity to explore related literature.

Answer: We think that the list of reference papers is made properly, helpful for explore of the related literatures. In the revised version, three reference papers are added.

Reviewer 4 Report

Comments and Suggestions for Authors

Dear Authors, thank you for submitting the paper titled “Ultrasonography and magnetic resonance imaging of bilateral ocular disease in a heifer”. It is an interesting case report, but at the current state the structure of the paper is not suitable for publication, and I would advise careful revision. Furthermore, there is no final diagnosis despite all the analysis performed, I think the readers would expect at least a tentative one.

The current structure of the paper is confusing; you present multiple healthy animals in the materials and methods, mixing some results there, then a single case report, then a discussion. If the current paper is a case report, please include only the diseased case; if it’s a study, then there must be a result section where you described the healthy animals and the diseased one.

As the healthy ones do not appear to have any role, I would consider delete them.

Materials and methods

Lines 76-78 and 79-84: these data belong to the results section. Why did you use three and then 4 animals? Were they (partially) the same - 3+1 - or different ones?

Lines 86-90: in which planes were the images acquired? transverse? Please specify. How many animals? Only the diseased one?

Case presentation

Lines 155-156: MRI post-mortem examination

Line 158: the structure could NOT be seen, but then you described it?

There is a detail missing in this section: what is your diagnosis? Do you believe this was just a different manifestation of the same described hereditary disease? A new one? Could some of the changes be acquired?

Discussion

I would start with the description of the diagnosed disease in your case, then comparison with the few other cases you described in the M&M, then comparison with the literature. I would use this scheme for the M&M and results as well (that would not be pertinent if you decide to present it as a case report, which I believe is the best option given the lack of utility of the information regarding the healthy patients).

In general, it is not very clear in the discussion when you refer to your case and when you are citing the literature. Please clearly describe your finding and then compare them with the pertinent literature.

Lines 286-291: these lines are not easy to read, and one doesn’t understand if you are talking about one report, two studies, your studies. I would consider re-writing.

Line 314: did you perform Doppler on your case?

Line 339: in this case could appear? Either it appeared or not.

Line 354: one bovine case? The bovine species in general?

Line 383: is this reference (27) comparing the utility of US, CT and MRI for ocular lesions?

Comments on the Quality of English Language

The quality of English is good in the introduction and case presentation, less in the discussion section where the use of tenses is not always correct and impairs the clearness of the concepts. 

Author Response

Dear Authors, thank you for submitting the paper titled “Ultrasonography and magnetic resonance imaging of bilateral ocular disease in a heifer”. It is an interesting case report, but at the current state the structure of the paper is not suitable for publication, and I would advise careful revision. Furthermore, there is no final diagnosis despite all the analysis performed, I think the readers would expect at least a tentative one.

Thank you for your providing important suggestion. In the revised version, the values obtained from the healthy, age-matched animals are deleted. In terms of final diagnosis for this case, based on the imaging results, we could suspect strongly that this case had the congenital (or hereditary) ocular diseases resembling with JB-MOD. However, unfortunately, our results were not enough for definitive diagnosis of a specific ocular disease including JB-MOD, because there were little differences between this case and the previous cases with various ocular diseases (including JB-MOD) in the symptom and imaging results. Thus, we agree that the content of this paper caused slightly tentative impression. Through the results of this paper, despite there is the difficulty in definitive diagnosis of bovine ocular diseases, we believe that this paper can provide importance of the combination uses of various imaging modalities for differential diagnosis of bovine ocular diseases, and the clinical trial may allow further detection of the unknown congenital or hereditary ocular diseases.

Question: The current structure of the paper is confusing; you present multiple healthy animals in the materials and methods, mixing some results there, then a single case report, then a discussion. If the current paper is a case report, please include only the diseased case; if it’s a study, then there must be a result section where you described the healthy animals and the diseased one. As the healthy ones do not appear to have any role, I would consider delete them.

Answer: We agree that the structure of our paper is not suitable for the text structure of case report. In the revised version, the values obtained from the healthy, age-matched animals are deleted. According to the deletion of the values from age-matched, healthy, Japanese-Black calves, the descriptions of the ultrasonographic measurements such as LMLs and APDs, and IOP are also deleted. Additionally, section of “Materials and Methods” is deleted.

Question: Lines 76-78 and 79-84: these data belong to the results section. Why did you use three and then 4 animals? Were they (partially) the same - 3+1 - or different ones?

Answer: In the revised version, section of “Materials and Methods” is deleted by transfer of the descriptions of ocular ultrasonography, MRI and PCR examination into the section “Case presentation”.

Question: Lines 86-90: in which planes were the images acquired? transverse? Please specify. How many animals? Only the diseased one?

Answer: MRI was carried out for the affected calf (not for healthy calves). In MRI examination, saggital, dorsal and transverse planes were acquired. In the present paper, the dorsal planes are used, because of more explainable than saggital and transverse planes.

Question: Lines 155-156: MRI post-mortem examination

Answer: According to this suggestion, this sentence is corrected.

Question: Line 158: the structure could NOT be seen, but then you described it?

Answer: Thak you for your providing important suggestion. This sentence included incorrect description. Thus, in the revised version, this sentence is corrected.

Question: There is a detail missing in this section: what is your diagnosis? Do you believe this was just a different manifestation of the same described hereditary disease? A new one? Could some of the changes be acquired?

Answer: The present results from these examinations were suggestive of a congenital multiple ocular disease, but not diagnostic in this case. Although this case might have involved with a new congenital or hereditary multiple ocular disease, we think that there is lack of evidence for the description about detection of new ocular disease. In the last part of section “Case presentation”, new sentence is added.

Question: I would start with the description of the diagnosed disease in your case, then comparison with the few other cases you described in the M&M, then comparison with the literature. I would use this scheme for the M&M and results as well (that would not be pertinent if you decide to present it as a case report, which I believe is the best option given the lack of utility of the information regarding the healthy patients).

Answer: The revised version would be suitable for the structure of case repot by deletion of “Materials and Methods” section.

Question: In general, it is not very clear in the discussion when you refer to your case and when you are citing the literature. Please clearly describe your finding and then compare them with the pertinent literature.

Answer: The descriptions of section “Discussion” are modified, with consideration for this suggestion.

Question: Lines 286-291: these lines are not easy to read, and one doesn’t understand if you are talking about one report, two studies, your studies. I would consider re-writing.

Answer: In the revised version, the sentence “The ultrasonographic appearances of the congenital ocular abnormalities … although these findings have not been shown in these two reports [5,6]” is deleted, and the sentence is modified.

Question: Line 314: did you perform Doppler on your case?

Answer: Doppler ultrasonography has not been applied for this case. Thus, this description is added in the revised version.

Question: Line 339: in this case could appear? Either it appeared or not.

Answer: Before this description in line 339, the description about visibility of the smaller structures such as V-shaped and corn-like structures of the detached retina is present.

Question: Line 354: one bovine case? The bovine species in general?

Answer: To our best knowledge, there was one previous bovine report describing contrast brain MRI, although some papers were published as describing about utility of contrast MRI for caprine case (i.g., Vet Radiol Ultrasound 2014;55:68-73), and contrast CT for the affected brain of the bovine cases (i.g., J Vet Med Sci 2011;73:113-115; Acta Vet Scand 2017;59:8).

Question: Line 383: is this reference (27) comparing the utility of US, CT and MRI for ocular lesions?

Answer: Reference paper (27) included comparison between ultrasonography and CT for visibility of soft tissue lesion in bovine abdomen (peritoneal mesothelioma). However, because this paper (27) is deleted from the reference paper in the revised version, because it did not include description about ocular lesions.

Question: The quality of English is good in the introduction and case presentation, less in the discussion section where the use of tenses is not always correct and impairs the clearness of the concepts.

Answer: The revised version of our paper includes great corrections in section “Discussion”.

Round 2

Reviewer 4 Report

Comments and Suggestions for Authors

Dear Authors, thank you for resubmitting this interesting case report. The current version has largely improved and I find it suitable for publication. 

Author Response

Dear Authors, thank you for resubmitting this interesting case report. The current version has largely improved and I find it suitable for publication. 

Thank you for your kindly suggestions for our manuscript.